# Synthesis and Properties of Nitrogen-Doped Carbon Quantum Dots Using Lactic Acid as Carbon Source

**DOI:** 10.3390/ma15020466

**Published:** 2022-01-08

**Authors:** Kaixin Chang, Qianjin Zhu, Liyan Qi, Mingwei Guo, Woming Gao, Qinwei Gao

**Affiliations:** 1College of Chemical Engineering, Nanjing Forestry University, Nanjing 210037, China; ckx851127@gmail.com (K.C.); 18260077835@163.com (Q.Z.); 18362985223@163.com (L.Q.); guomw199876@163.com (M.G.); orangtreechina@gmail.com (W.G.); 2Jiangsu Co-Innovation Center of Efficient Processing and Utilization of Forest Resources, Nanjing Forestry University, Nanjing 210037, China

**Keywords:** nitrogen-doped carbon quantum dots, fluorescence, pH value, Fe^3+^ ion, lactic acid

## Abstract

Nitrogen-doped carbon quantum dots (N-CQDs) were synthesized in a one-step hydrothermal technique utilizing L-lactic acid as that of the source of carbon and ethylenediamine as that of the source of nitrogen, and were characterized using dynamic light scattering, X-ray photoelectron spectroscopy ultraviolet-visible spectrum, Fourier-transformed infrared spectrum, high-resolution transmission electron microscopy, and fluorescence spectrum. The generated N-CQDs have a spherical structure and overall diameters ranging from 1–4 nm, and their surface comprises specific functional groups such as amino, carboxyl, and hydroxyl, resulting in greater water solubility and fluorescence. The quantum yield of N-CQDs (being 46%) is significantly higher than that of the CQDs synthesized from other biomass in literatures. Its fluorescence intensity is dependent on the excitation wavelength, and N-CQDs release blue light at 365 nm under ultraviolet light. The pH values may impact the protonation of N-CQDs surface functional groups and lead to significant fluorescence quenching of N-CQDs. Therefore, the fluorescence intensity of N-CQDs is the highest at pH 7.0, but it decreases with pH as pH values being either more than or less than pH 7.0. The N-CQDs exhibit high sensitivity to Fe^3+^ ions, for Fe^3+^ ions would decrease the fluorescence intensity of N-CQDs by 99.6%, and the influence of Fe^3+^ ions on N-CQDs fluorescence quenching is slightly affected by other metal ions. Moreover, the fluorescence quenching efficiency of Fe^3+^ ions displays an obvious linear relationship to Fe^3+^ concentrations in a wide range of concentrations (up to 200 µM) and with a detection limit of 1.89 µM. Therefore, the generated N-CQDs may be utilized as a robust fluorescence sensor for detecting pH and Fe^3+^ ions.

## 1. Introduction

Carbon quantum dots (CQDs) are regarded as a novel class of fluorescence nanomaterials. Meanwhile CQDs exhibit many unique advantages, compared with traditional fluorescent nanomaterials, such as excellent biocompatibility [1], good solubility [2], low toxicity [3,4], remarkable photostability [5], as well as their small size. Hence CQDs are considered as one of the most attractive alternative to traditional fluorescent materials, and have been used in drug delivery carriers [6,7], biological imaging [8,9], fluorescent probes [10,11], photovoltaic devices [12,13], and so on.

CQDs were originally founded by Xu et al. [14] in 2004, who utilized an electrophoretic separation and purification technique for the purification of single-walled carbon nanotubes generated from arc-discharge soot, and named as carbon quantum dots by Su et al. [15] in 2006. Over the past decade, many synthetic methods of CQDs have been reported, such as arc-discharge [14], laser ablation [16], electrochemical oxidation [17], microwave method [18], and hydrothermal treatment [19]. These synthetic methods have two main purposes: one purpose is to crack different carbon source materials, and the other is the carbonization of small molecules or polymers [20]. Meanwhile, a variety of carbon-sources materials, including graphite [21], citric acid [22,23], watermelon peel [24], saccharum officinarum juice [25], scindapsus leaves [26], and others, have been developed for the preparation of CQDs. Therefore, various synthesis methods and carbon sources may obtain CQDs with various sizes, structures and excitation wavelengths, which will have an obvious impact on the fluorescence emission intensity of CQDs. The two disadvantages including low yield of CQDs and a small amount of functional groups on CQD surface limit the fluorescence performance and application of CQDs [27]. When CQDs are doped by nitrogen, sulfur, and other elements, their spectral properties and applications may be improved. Thus N-doped carbon quantum dots (N-CQDs) and S-doped carbon dots (S-CQDs) can be obtained, while the doping of N and S elements may change the electron density, structure, and composition of the doped CQDs, leading to the increase of fluorescence intensity of N-CQDs and S-CQDs [28,29], which will boost CQDs performance and broaden CQDs application. Recently, much research has reported on the N-doped and N/other atom co-doped CQDs. The N-doped CQDs are widely used in the field of probes, especially for metal ions detection by probes. Either in nature or in animals, iron is an important element, especially Fe^3+^ ions. Therefore, it is essential to detect Fe^3+^ ions. Qi et al. [30] utilized a one-step hydrothermal approach to manufacture nitrogen-doped carbon quantum dots N-CQDs with rice residue as carbon source and glycine as nitrogen source, respectively, and then used N-CQDs as the probe to detect Fe^3+^ ions and tetracycline antibiotics. Du et al. [31] adopted hydrothermal technique using ascorbic acid and thioglycolic acid to synthesize sulfur-doped CQDs, which were used as the fluorescent probe to detect Fe^2+^ and Fe^3+^ ions in oral ferrous gluconate samples. Chen et al. [32] synthesized N-CQDs by hydrothermal method with p-phenylenediamine and ammonia, and the N-CQDs were considered as a multi-functional fluorescence sensor for detecting pH and Fe^3+^ ions.

L-lactic acid is a widely used bio-based chemical containing hydroxyl and carboxyl groups, while ethylenediamine contains amino groups, which may be good candidates for preparing N-CQDs. In this work, we used a one-step hydrothermal technique to create N-CQDs utilizing L-lactic acid as carbon source and ethylenediamine as nitrogen source, respectively. The obtained N-CQDs featuring surface functional groups present excellent water solubility and fluorescence. The N-CQDs were investigated in terms of their morphology, surface structure, and optical properties. Those N-CQDs have been synthesized that were spherical particles with overall diameters ranging from 1–4 nm. When irradiated by ultraviolet light with the wavelength of 365 nm, N-CQDs would emit blue light, and their emission wavelength reaches to the maximum of 414 nm during the excitation wavelength of ultraviolet light being 310 nm, which indicates that the optical characteristics of N-CQDs depend on the excitation wavelength. Meanwhile, the fluorescence of N-CQDs also reveals a pH-dependent dependency. Furthermore, the obtained N-CQDs showed better specificity and sensitivity to Fe^3+^ ions, suggesting that they might be exploited as fluorescent probe of Fe^3+^ ions. The research of N-CQDs prepared from lactic acid would expand the scope of application of lactic acid.

## 2. Materials and Methods

### 2.1. Materials

L-lactic acid (A.R., 90%) was purchased from Musashino Chemical (Yichun, China) Corporation. Ethylenediamine (A.R.), hydrochloric acid (A.R.), NaOH (A.R.), FeCl_3_ (A.R.), FeCl_2_·4H_2_O (A.R.), CuCl_2_·2H_2_O (A.R.), MgCl_2_·6H_2_O (A.R.), NiCl_2_·6H_2_O (A.R.), CaCl_2_ (A.R.), SnCl_2_ (A.R.), KCl (A.R.), CoCl_2_·6H_2_O (A.R.), and LiCl·2H_2_O (A.R.) were all bought from Nanjing Chemical Reagent Company (Nanjing, China). A dialysis bag with a molecular weight cut-off of 1000 Da was purchased from Shanghai Yuanye Bio-Technology Co., Ltd (Shanghai, China). The deionized water was purified using an Ulupure system (Nanjing Youpu Environmental Protection Equipment Co., Ltd, Nanjing, China) throughout all the experiments. All reagents used were available commercially and did not involve purification.

### 2.2. Synthesis of Nitrogen-Doped Carbon Quantum Dots

We synthesized N-CQDs through one-step hydrothermal method employing L-lactic acid as carbon source and ethylenediamine as nitrogen source [26,33] (Figure 1). 30.0 g L-lactic acid and 2 mL ethylenediamine were mixed with 30 mL deionized water in a 100 mL three-necked flask, then the solution was heated up to 150 °C and reacted for 12 h. The reaction solution was then allowed to cool naturally before even being centrifuged for 10 min at 10,000 rpm, The supernatant was collected and transferred to a dialysis bag with a molecular weight cut-off of 1000 Da, then dialyzed for 72 h with deionized water in the glass flume. In the end, the dialyzed supernatant was freeze-dried in vacuum freeze dryer to yield solid N-CQDs.

In this section, the LG16-A high-speed centrifuge (Lab Centrifuge, Beijing, China) was used to centrifuge the sample to eliminate big particles of contaminants, while the FD-1A-50 freeze drier (Boyikang Experimental Instrument Co., Ltd., Beijing, China) was used for freeze drying to create N-CQDs solids.

### 2.3. Characterization Methods

The UV-vis spectrophotometer UV-2450 (Shimadzu, Kyoto Japan) was applied to detect UV-vis spectra. 0.05 mg/mL N-CQDs aqueous solution prepared by dispersing the solid N-CQDs in deionized water was used for both UV-vis and fluorescence spectra below. Fourier-transformed infrared spectroscopy (FTIR) was measured with the FTIR-360 (PerkinElmer, Waltham, MA, USA). The scanning range for the FTIR spectra was 450–4500 cm^−1^, and the KBr tablet was being used as a carrier. The microstructures of N-CQDs were examined using JEM-2100 (JEOL, Tokyo, Japan) High-resolution transmission electron microscopy (HRTEM). To determine the elemental composition, X-ray photoelectron spectroscopy (XPS) using an Al Kα monochromatized source was studied on AXIS UltraDLD (Kratos, Kyoto, United Kingdom). Fluorescence spectra were recorded on Perkinelmer Fluorescent FL6500 (PerkinElmer, Waltham, Massachusetts, USA). All fluorescence spectra were measured with the excitation slit being 5 nm, the emission slit being 10 nm, and the scanning speed being 1200 nm/min. Zetasizer Nano-ZS (Malvern instruments, Malvern, UK) was being used to evaluate the size and distribution of N-CQDs nanoparticles.

### 2.4. Quantum Yield of Nitrogen-Doped Carbon Quantum Dots

Quinine sulfate dissolved in 0.1 M sulfuric acid is used as the standard sample, and its quantum yield (*Q_S_*) is 0.54. The quantum yield of N-CQDs (*Q_N_*) is calculated that used the following equation [34,35].
(1)$$Q_{N}=Q_{S}\frac{I_{N}}{I_{S}}\frac{A_{S}}{A_{N}}\frac{\varphi_{N}^{2}}{\varphi_{S}^{2}}$$
where *Q* stands for the quantum yield and I for the integrated area of fluorescence intensity, while *A* for the absorbance and *φ* for the refractive index of the solvent (being 1.33). The standard sample and N-CQDs are denoted by the subscripts *S* and *N*, respectively.

### 2.5. Effect of pH Value on the Fluorescence Intensity of N-CQDs

A certain amount of N-CQDs solid was dissolved in deionized water to yield 0.05 mg/mL N-CQDs solution. Aqueous solutions with pH values being from 1 to 14 were prepared with hydrochloric acid and sodium hydroxide, respectively. 1 mL of N-CQDs solution and 1 mL hydrochloric solution (or NaOH solution) with different pH values were added to a quartz cuvette and mixed for 1 min, and then the fluorescence test was performed at ambient temperature. All fluorescence spectra were recorded at 310 nm for excitation, and 414 nm for emission.

### 2.6. Fluorescence Detection of Fe^3+^ Ions

The above-mentioned aqueous solution of N-CQDs had a concentration of 0.05 mg/mL. The chlorides containing various metal ions (including Li^+^, Ca^2+^, Co^2+^, Cu^2+^, Fe^3+^, Mg^2+^, Fe^2+^, Ni^2+^, Sn^2+^ and K^+^) were, respectively, dissolved in deionized water to prepare various chloride solutions with the concentration of 1 mM. To investigate the selectivity of N-CQDs aqueous solution to various metal ions, 0.5 mL chloride solution containing different metal ions and N-CQDs aqueous solution with a volume of 2 mL were added to a quartz cuvette and mixed at ambient temperature for 1 min before even being detected. To evaluate the detection range of Fe^3+^ ions and the sensitivity of N-CQDs to Fe^3+^ ions, 0.5 mL of Fe^3+^ solutions with their concentrations varying from 0 μM to 200 μM were mixed with N-CQDs aqueous solution with a volume of 2 mL at ambient temperature for 2 min, and then the mixed solutions were detected. 0.5 mL FeCl_3_ solution and 0.5 mL other chloride solution were mixed with N-CQDs aqueous solution with a volume of 2 mL, and then the mixture solutions were detected to determine the fluorescence quenching of N-CQDs triggered by Fe^3+^ together with other metal ions.

## 3. Results

### 3.1. The Morphology and Surface Composition of N-CQDs

The N-CQDs were characterized using HRTEM to determine their morphology and surface functional groups. The HRTEM images of N-CQDs (Figure 1a) display that N-CQDs are spherical in morphology with diameters ranging from 1–4 nm (Figure 1b), which is consistent with prior literatures [36].

An FTIR test was conducted to analyze the functional groups on the surface of N-CQDs. The FTIR spectrum of N-CQDs in Figure 1c demonstrates a peak at 1447 cm^−1^ corresponding to the stretching vibration band of the amine C–N bond [37], and the peak at 3292 cm^−1^ owing to the stretching vibration bands of O–H and N–H [38], indicating the existence of amino functional groups. Meanwhile, the stretching vibration of C=C is associated with the peak at 1540 cm^−1^ [39,40]. Stretching C–H of alkyl groups results in the two peaks at 2942 and 2985 cm^−1^ [35,41]. The peak at 1744 cm^−1^ owing to carboxyl C=O bonds indicates carbonyl groups in N-CQDs [42], whereas the peak at 1655 cm^−1^ due to amide C=O indicates the amide groups of N-CQDs [37]. the stretching vibration of C–O of carbonyl groups causes the peak at 1127 cm^−1^ [43]. The FTIR results clearly illustrate that nitrogen was effectively doped into the CQDs and are consistent with those of XPS below.

The surface chemical composition and element status of N-CQDs were investigated further utilizing X-ray photoelectron spectroscopy (XPS). The XPS spectrum of N-CQD in Figure 2a shows three peaks at 532.3 eV, 286 eV and 400.4 eV, corresponding to O1s, C1s and N1s, respectively [26,44,45], indicating that N-CQDs are mostly constituted of carbon, nitrogen, and oxygen. The high resolution XPS spectrum of C1s in Figure 2b shows five peaks at 284.6 eV, 285.5 eV, 286.4 eV, 287.1 eV and 288.3 eV that may be traced to C=C, C–C, C–N, C–O and C=O bonding on the surface of N-CQDs [26,46]. In Figure 2c, the high resolution XPS spectrum of N1s displays two peaks at 401.3eV and 399.8 eV caused by C–N and N–H, respectively [32,47]. The high resolution XPS spectrum of O1s in Figure 2d exhibits two peaks at 531.5 eV and 532.4 eV, which may be traced to C–O and C=O, respectively [48,49]. The results of XPS, together with those of HRTEM and FTIR show that N-CQDs were effectively fabricated. Moreover, N-CQDs samples possess carboxyl, hydroxyl and amino groups, resulting in N-CQDs’ excellent water solubility.

### 3.2. Optical Performance of N-CQDs

The UV-Vis absorption spectrum was used to characterize the optical property of the generated N-CQDs. The UV–Vis absorption spectrum of N-CQDs aqueous solution in Figure 3a exhibits two absorbance bands severally centered at 220 nm and 320 nm. The peak at 220 nm is ascribable to the π-π* transition for C=C bond [32], while the peak at 320 nm is owing to the n-π* transition of C=O bond or surface defects of N-CQDs [36]. As seen in the inset in Figure 3b, the aqueous solution of N-CQDs was yellow in visible light and emitted blue fluorescence with the wavelength of 365 nm in UV light.

Figure 3c displays that the N-CQD samples prepared from L-lactic acid possess very good fluorescence properties. Their fluorescence intensity will be affected by excitation wavelength. Their fluorescence intensity steadily declines as the excitation wavelength increases from 300 nm to 380 nm with each increment of 10 nm. The normalized fluorescence spectra of N-CQDs given in Figure 3d show that the fluorescence emission peak of N-CQDs would red-shift by 22 nm from 414 nm to 436 nm with the excitation wavelength changing from 300 nm to 380 nm. This red shift may be caused by the varied sizes of N-CQDs as well as the functional groups on the surface of the N-CQDs [36,50]. Moreover, N-CQDs possess a maximal emission peak at 414 nm with an excitation wavelength of 310 nm. Meanwhile, the quantum yield of N-CQDs is 46%, which is significantly higher than those of the CQDs synthesized from biomass as carbon source in literatures shown in Table 1.

### 3.3. The Effect of pH on N-CQDs Fluorescence Intensity

The surface of the obtained N-CQDs includes amino, hydroxyl and carboxyl functional groups, as can be seen by XPS and FTIR. We may utilize these functional groups to detect the effect of pH values on N-CQDs fluorescence intensity. The fluorescence intensities of N-CQDs with different pH values at the excitation wavelength of 310 nm were given in Figure 4a,b, which clearly show the apparent effects of pH on N-CQDs fluorescence intensity. As shown in Figure 4c,d, when pH value increases from 1 to 7, N-CQDs fluorescence intensity gradually increases and reaches to the maximum as pH value being 7. Then the N-CQD fluorescence intensity decreases with rising pH values from 8 to 14 and reaches a minimum as pH value being 14. N-CQDs fluorescence intensity slightly vary as pH values from 3 to 12, but they are much lower under strong acid (pH being 1 or 2) or alkali conditions (pH being 13 or 14). In Figure 4c, N-CQDs fluorescence intensity is quenched under acidic conditions while the emission peak of N-CQDs does not change, which may be due to the protonation of amino groups on N-CQDs surface [32]. In Figure 4d, both fluorescence intensity and emission peak of N-CQDs diminish as pH increases, which may be attribute to the protonation of carboxyl groups on the surface of N-CQDs [43,57].

In short, the pH-dependent fluorescent behavior of N-CQDs might be ascribed to functional groups on the surface of N-CQDs. When the surface functional groups of N-CQDs are protonated, fluorescence quenching occurs mainly due to the agglomeration of N-CQDs particles [58]. Therefore, N-CQDs can be developed for the use in the field of pH probes based on its sensitivity to pH.

### 3.4. Detection and Selection of Fe^3+^ Ions by N-CQDs

The hydroxyl, carboxyl and amino surface functional groups of N-CQDs can form complexes with metal ions [59]. As the number of functional groups on the surface of the N-CQDs decreases, its fluorescence intensity would also gradually weaken. Therefore, it may be utilized as a probe to detect metal ions. In this work, to investigate specificity and sensitivity of the obtained N-CQDs to various metal ions, we measured the fluorescence intensities of N-CQDs aqueous solution complexed by different metal ions with the concentration of 1 mM under the same condition. Figure 5a,b demonstrated the influence of metal ions on N-CQDs fluorescence intensity at 310 nm excitation wavelength. Although each metal ion has different effect on N-CQDs fluorescence intensity, this can lessen its fluorescence intensity. The fluorescence quenching performance of Fe^3+^ ions for N-CQDs is obviously superior to that of other metal ions, as N-CQDs fluorescence intensity would decrease by about 99.6% due to Fe^3+^ ions, indicating that they have quite an excellent selectivity for Fe^3+^. To detect the fluorescence quenching of N-CQDs induced by Fe^3+^ in combination with other metal ions, a mixture of Fe^3+^ ions with other metal ions was added to an aqueous solution of N-CQDs, and then the mixed solutions were detected. Figure 5c exhibited that the effect of Fe^3+^ ions on N-CQD fluorescence quenching is little affected by other metal ions. These findings suggest that they have a specific recognition effect on Fe^3+^ and a strong anti-interference ability.

In order to make sure the sensitivity of N-CQDs to Fe^3+^ ions, the effect of Fe^3+^ ions concentrations on N-CQDs fluorescence intensity at 310 nm excitation wavelength was shown in Figure 5d. Its fluorescence intensity steadily declines with Fe^3+^ concentration increasing from 0 μM to 200 μM. Figure 5e displays a good linearity between the quenching efficiency (i.e., the difference (*F_0_* − *F*)) and Fe^3+^ concentrations ranging from 25 to 175 μM with the correlation coefficient of 0.99602. Thus, the concentration of Fe^3+^ ions could be computed using the equation below.
$$F_{0}-F=470.657C+100297.143$$
where *F* and *F_0_* denote the fluorescence intensities of N-CQDs in the presence and the absence of Fe^3+^, respectively. The concentration of Fe^3+^ ions is represented by *C*. According to the above equation, the limit of detection is calculated to be 1.89 μM based on three times the standard deviation rule (3δ/k, where δ represents standard deviation of the blank N-CQDs and k is the slope of the calibration curve, *n* = 6), and it is far below the requirement that Fe^3+^ ion concentration in drinking water is less than 5.4 μM. Table 2 compares the detection capability of the obtained N-CQDs for Fe^3+^ to that of CQDs for Fe^3+^ reported in other literatures, in which the linear range of Fe^3+^ detection was ordered from small to large range. The obtained N-CQDs have quite a broad linear detection range and are far more sensitive to Fe^3+^ ions.

The complexation of Fe^3+^ with the carboxyl group, amino group and hydroxyl group on the surface of N-CQDs may be responsible for the fluorescence quenching caused by Fe^3+^ [26,60]. Meanwhile, the N atom has a large electronegativity, which is conducive to enhancing the electron density distribution on N-CQDs surface and promoting the complexation of Fe^3+^ with N-CQDs surface functional groups [61]. The complexation will facilitate the rapid movement of electrons between Fe^3+^ ions and N-CQDs, and lead to the formation of a non-radiative electron/hole recombination [47]. Therefore, Fe^3+^ ions significantly quench the fluorescence of N-CQDs. Furthermore, the particle size of N-CQDs significantly increases with the addition of Fe^3+^ ions, for the diameter distributing of N-CQDs increases from 2.6 nm to 62.3 nm (in Figure 5f), and the green curve in Figure 5f is obtained by fitting Gaussian function to the particle size of N-CQDs solution after adding Fe^3+^, which proves that Fe^3+^ ions complexed with the surface functional groups of N-CQDs. All results show high selectivity and sensitivity of the obtained N-CQDs to Fe^3+^ ions.

**Table 2 materials-15-00466-t002:** Comparison on CQDs for determination of Fe^3+^ ions.

Materials	Method	Linear Range (µM)	Limit of Detection (µM)	Reference
Sulfanilic acid	solvothermal	0.025–0.4	2.549	[62]
Isoleucine and citric acid	hydrothermal	0–20	-	[60]
Roasted chickpea	Microwave	11.25–37.5	2.74	[63]
Rice residue and lysine	hydrothermal	3.32–32.26	0.7462	[64]
Citric acid and Tris	hydrothermal	2–50	1.3	[65]
L-glutamic acid and ethylenediamie	microwave	8–80	3.8	[33]
Wheat straw	hydrothermal	0–250	1.95	[66]
Lactic acid and ethylenediamie	hydrothermal	25–175	1.89	This work

## 4. Conclusions

In summary, we had been using a one-step hydrothermal technique to create N-CQDs utilizing L-lactic acid as a carbon source and ethylenediamine as a nitrogen source, respectively. The N-CQDs created are nanospheres with outstanding fluorescence and good solubility in water, while the quantum yields of N-CQDs (being 46%) are significantly higher than that of the CQDs reported in literatures. Furthermore, the emission intensity of N-CQDs exhibits excitation wavelength dependence. Their fluorescence intensity steadily decreases as the excitation wavelength increases from 310 to 380 nm. Meanwhile, the fluorescence intensity of N-CQDs shows sensitivity to pH values. In comparison to other metal ions, N-CQDs have quite a high specificity and sensitivity for Fe^3+^ ions in a wide range of concentrations with the low detection limit of 1.89 µM. Therefore, the N-CQDs created might be employed as fluorescent probe for pH and Fe^3+^ ions.

## Data Availability

The data presented in this study are available on request from the corresponding author.

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
