# Peer review of "Synthesis and Properties of Nitrogen-Doped Carbon Quantum Dots Using Lactic Acid as Carbon Source"

_materials, 2022, doi:10.3390/ma15020466_

Round 1

Reviewer 1 Report

Good work with proven results. The manuscript deserves to be published in the Journal. But even so, I would like to give some minor questions/comments.

In the Introduction, the transition from CQD to N-CQD is made too abrupt. For a better overall impression, you need to either review CQD less or more N-CQD.

All figures, especially the Graphical abstract, are of very low resolution! Please resubmit it in hires .pdf or .eps file type.

P.1. L.14. For a wider readership, “particle size analyzer” should be replaced with “dynamic light scattering”. Hereinafter in the text.

P.4. L.109. “…KBr tablet was being used as a carrier” sounds better.

P.7. L.191-192. Sentence verification; “as”; “The”.

Table 1. What is the principle of the rows’ order? Neither by source name, nor by method, nor by QY, nor by reference number. It is difficult to follow. May be a presentation, for example, by percentage growth will be more informative. It is acceptable that the results of this work were presented in the last row.

Table 2. Again about data representation in a frame of an order of the table rows. Some criteria should be. For example by range or detection limit.

Fig 5f. What means green solid line? Approximation? Not the best fit. The experimental pattern is quite symmetrical but the fit isn’t. Please explain.

Author Response

We thank the reviewers very much for their valuable comments. We have revised the manuscript thoroughly according to the reviewers' comments and explained point-by-point the details of the revisions in the manuscript and our responses to the reviewers' comments. Please see the attachment.

Reviewer 2 Report

The manuscript presents the synthesis and characterization of nitrogen-doped carbon quantum dots using lactic acid as carbon source and ethylenediamine as nitrogen source. The obtained nitrogen-doped carbon quantum dots were characterized using a particle size analyzer, X-ray photoelectron spectroscopy ultraviolet-visible spectrum, Fourier-transformed infrared spectrum, high-resolution transmission electron microscopy, and fluorescence spectrum.

The synthetized N-CQDs showed significant fluorescence quenching at pH lower or higher then pH 7 and also in presence of Fe3+ ions. The authors proposed the utilization of the synthetized N-CQDs as a fluorescence probe for detecting pH and Fe3+ ions.

The study was well designed and the results are promising for application of the prepared nitrogen-doped carbon quantum dots that showed fluorescence properties in the analysis of pH and Fe3+ ions.

I recommend the following corrections:

The punctuation of the manuscript must be carefully revised.

In section 2.1 Materials the quality and the supplier of the substances used must be indicated and also the equipment used for centrifugation, dialysis and vacuum freeze dryer.

Author Response

(The authors gave the same response as above.)

Reviewer 3 Report

  1. Materials - please provide the suppliers and all equipment as used

  2. Please provide an example for calculations of The quantum yield of N-CQDs.

  3. Figure 1 should have better resoilution, especially HRTEM images.

  4. All figures should have better resolution!

  5. There are numerous typos and grammar mistakes, e.g., Fig.3(c) display; possess very good fluorescence properties.; to synthesis sulfur; below.[34, 35]...etc.

  6.  What is new in your contribution? please provide motivation

Author Response

(The authors gave the same response as above.)

Round 2

Reviewer 3 Report

The manuscript can be published in the present form.